# Scorpion Venom as a Source of Antimicrobial Peptides: Overview of Biomolecule Separation, Analysis and Characterization Methods

**DOI:** 10.3390/antibiotics12091380

**Published:** 2023-08-29

**Authors:** Sara Nasr, Adolfo Borges, Christina Sahyoun, Riad Nasr, Rabih Roufayel, Christian Legros, Jean-Marc Sabatier, Ziad Fajloun

**Affiliations:** 1Laboratory of Applied Biotechnology (LBA3B), Azm Center for Research in Biotechnology and Its Applications, EDST, Lebanese University, Tripoli 1300, Lebanon; sara.nasr99@gmail.com (S.N.); christina.sahyoun@etud.univ-angers.fr (C.S.); 2Laboratorio de Biología Molecular de Toxinas y Receptores, Instituto de Medicina Experimental, Facultad de Medicina, Universidad Central de Venezuela, Caracas 50587, Venezuela; borges.adolfo@gmail.com; 3Centro para el Desarrollo de la Investigación Científica, Asunción 1255, Paraguay; 4Univ Angers, INSERM, CNRS, MITOVASC, Team 2 CarMe, SFR ICAT, 49000 Angers, France; 5Department of Physical Therapy, Faculty of Public Health 3, Lebanese University, Tripoli 1200, Lebanon; riadnasr1974@gmail.com; 6College of Engineering and Technology, American University of the Middle East, Egaila 54200, Kuwait; rabih.roufayel@aum.edu.kw; 7Aix-Marseille Université, CNRS, INP, Inst Neurophysiopathol, 13385 Marseille, France; 8Faculty of Sciences 3, Department of Biology, Lebanese University, Campus Michel Slayman Ras Maska, Tripoli 1352, Lebanon

**Keywords:** scorpion venom, antimicrobial peptides, proteomic, separation techniques, analytical methods, RP-HPLC, bioactive molecules, size exclusion chromatography, electrophoresis

## Abstract

Scorpion venoms have long captivated scientific researchers, primarily due to the potency and specificity of the mechanism of action of their derived components. Among other molecules, these venoms contain highly active compounds, including antimicrobial peptides (AMPs) and ion channel-specific components that selectively target biological receptors with remarkable affinity. Some of these receptors have emerged as prime therapeutic targets for addressing various human pathologies, including cancer and infectious diseases, and have served as models for designing novel drugs. Consequently, extensive biochemical and proteomic investigations have focused on characterizing scorpion venoms. This review provides a comprehensive overview of the key methodologies used in the extraction, purification, analysis, and characterization of AMPs and other bioactive molecules present in scorpion venoms. Noteworthy techniques such as gel electrophoresis, reverse-phase high-performance liquid chromatography, size exclusion chromatography, and “omics” approaches are explored, along with various combinations of methods that enable bioassay-guided venom fractionation. Furthermore, this review presents four adapted proteomic workflows that lead to the comprehensive dissection of the scorpion venom proteome, with an emphasis on AMPs. These workflows differ based on whether the venom is pre-fractionated using separation techniques or is proteolytically digested directly before further proteomic analyses. Since the composition and functionality of scorpion venoms are species-specific, the selection and sequence of the techniques for venom analyses, including these workflows, should be tailored to the specific parameters of the study.

## 1. Introduction

Scorpion stings are common in tropical and subtropical regions, with an estimated 1.2 million stings per year and over 2600 deaths [1]. In nature, scorpions use the stinger in their tail to inject venom into predators as a defense mechanism or into preys to paralyze and capture them. Scorpion venoms are heterogeneous mixtures composed of salts, free amino acids, varying concentrations of proteins (including neurotoxins and neurotoxin-related molecules such as components with lipolytic activity), enzymes (including hyaluronidases, metalloproteinases, phospholipases and chitinases), and peptides (comprising antimicrobial components and bradykinin-potentiating peptides, among other components) [2,3,4,5,6]. Thus, upon envenomation, signs and symptoms may vary depending on neurotoxin concentration, with clinical manifestations ranging from local effects, such as pain and edema, to severe, life-threatening complications, mainly comprising neurological and cardiorespiratory alterations [7]. The current treatment for scorpion envenomation relies on the prompt administration of specific antivenoms. Hence, it is imperative to thoroughly explore and identify scorpion venom components, which often exhibit species-specific mechanisms of action [1,3]. This intensive investigation is crucial in order to facilitate the design of more effective immunotherapies. Another reason that solicits the in-depth study of scorpion venom components is their promising therapeutic potential. Namely, scorpion antimicrobial peptides (AMPs) exert anti-bacterial effects against some resistant bacterial strains, implying a potential role in the treatment of diseases associated with drug-resistant bacteria. Mucroporin-M1, Imcroporin and Vejovine are some AMPs that have shown strong anti-bacterial activity despite their hemolytic effect [8,9,10,11]. Scorpine, a disulfide-bridged peptide (DBP) from *Pandinus imperator* and two non-disulfide bridged-peptides (NDBPs), Meucin 24 and Meucin 25, from *Mesobuthus eupeus*, exhibited potent antimalarial (i.e., anti-*Plasmodium falciparum*) effects [12,13]. Chlorotoxin (CTx), a well characterized toxin derived from *Leiurus hebraeus* venom, is known for its high selectivity towards glioma cell lines without hampering normal cells [14]. Until now, the search for scorpion venom-derived active components that can be used for drug development is still in progress due to the undiscovered properties of venom components from many scorpion species worldwide. Originally focused on components derived from genera within the Buthidae family, which include species with highly lethal venoms for humans, novel venom molecules have been discovered from genera within the families Hormuridae, Chactidae, Scorpionidae, Chaerilidae, Urodacidae, Liochelidae, Diplocentridae, and Hemiscorpiidae, among others, with generally lesser medical importance (except for the genus *Hemiscorpius*), underscoring the biochemical diversity of scorpion venoms, which serve as extensive combinatorial libraries throughout this arachnid order [15]. 

Scorpion venoms are complex cocktails that contain neurotoxins, particularly prominent in buthid venoms, which are among the most promising, potentially therapeutic components due to their high selectivity towards various isoforms of voltage-gated ion channels [16]. Nonetheless, venom has arisen as a powerful tool to address different problems and develop solutions in many domains, especially pharmacological and medical [17]. Furthermore, because of the relatively small number of taxa whose venoms have been studied in relation to the total number of world scorpion species, only 14.9% of data on scorpions’ peptides and proteins are currently accessible. Thus, improvements in the separation of scorpion venom components are desirable, as it constitutes the first step to decipher and analyze the rich complexity of scorpion venoms [18,19]. In this review, we describe the main techniques used to separate, isolate, and characterize AMPs and other bioactive molecules present in scorpion venoms. Thus, we first detail the combinations of electrophoresis and chromatographic techniques used to purify and identify peptides and proteins from venom and then proceed to describe the implementation of separation methods for venom components. 

## 2. Methods Used for Separation of Venom Complex Mixtures

### 2.1. Electrophoretic Separation Techniques

#### 2.1.1. One-Dimension Gel Electrophoresis (1-DGE) 

1-DGE, also known as sodium dodecyl sulfate–polyacrylamide gel electrophoresis (SDS-PAGE), is the most used qualitative technique for assessing protein separation in scorpion venom analyses. The separated proteins appear as bands that are visualized by Coomassie Blue Brilliant (CBB) dye or silver staining for better visualization of low molecular mass components [20]. SDS-PAGE (generally 12–20% polyacrylamide gels) is greatly used to compare the general composition of scorpion venoms, particularly regarding inter- and intra-specific venom variations. Scorpion intra-species differences in protein content were revealed by SDS-PAGE for *Mesobuthus tamulus* venom deriving from scorpion populations inhabiting locations in western and southern India [21]. Similarly, another study evaluated intra-species variation of venoms obtained from *Leiurus quinquestriatus* collected from different geographical locations in Egypt. Their SDS-PAGE profiles indicated components with population-specific electrophoretic migration, thus revealing a difference in venom composition depending on geographical origin [22]. Furthermore, in scorpion venom analysis, SDS-PAGE can be coupled with different techniques to gain a better understanding of the scorpion venom composition. For instance, SDS-PAGE was used to follow up the separation of the constituents of *Mesobuthus eupeus* scorpion venom in immunogenic protein fractions by affinity chromatography, which led to determine that immunogenic components correspond to the major part of crude *M. eupeus* venom, refuting previous beliefs [23]. Moreover, *Tityus pachyurus*, *Tityus cerroazul*, *Tityus obscurus*, *Tityus perijanensis*, *Tityus discrepans*, *Tityus zulianus* and *Tityus serrulatus* venom proteins were each separated by SDS-PAGE also showing differences in venom protein composition within the genus *Tityus*. Immunoblotting can follow SDS-PAGE separation to compare venom antigenic reactivity to experimental or therapeutic antivenoms [24]. Reverse-phase high-performance liquid chromatography (RP-HPLC) fractionation prior to SDS-PAGE could help in narrowing and focusing the final result [20]. Additionally, SDS-PAGE can be employed in a series of steps for protein identification and characterization, which is commonly referred to as proteomics. SDS-PAGE can help to determine what fraction should be further tested depending on its molecular mass. The separation result will thus be followed directly by in-gel tryptic digestion of selected bands and a mass spectrometry analysis; or, for a better resolution and more sensitivity, RP-HPLC can precede the in-gel digestion [25]. SDS-PAGE can also be used as a final step to confirm the purification of a component from a venom mixture after several fractionation steps to assess the quality and purity of the specific separated molecule(s) prior to further analyses [26,27]. Notably, SDS-PAGE allows processing of a small number of proteins (less than 50) which in some cases can be a limitation [28].

#### 2.1.2. Two-Dimension Gel Electrophoresis (2-DGE)

Despite the simplicity and the high-resolution separation that is offered by SDS-PAGE, 2-DGE might be employed to analyze more complex protein mixtures with a higher resolution [29]. This method separates venom components based on two dimensions (molecular mass and isoelectric point (pI)). 2-DGE allows the identification of venom components by generating a spot map, where the densiometric intensity of each spot can help determine the relative abundance of proteins [30]. It provides data about both the molecular mass and the pI of the identified spot [27]. Hence, 2-DGE performed under identical conditions allows the comparison of venom from different taxa/geographical origins to evidence venom variability by comparing protein distribution [31]. This technique also aids in narrowing down the search for specific proteins that require further analysis and purification. In the purification process of the heterodimeric phospholipase A2 Heteromtoxin (HmTx) from *Heterometrus laoticus* venom (14,018.4 Da; pI 5.6), 2-DGE was conducted to identify four major groups of proteins that formed distinct spots with varying isoelectric points (pIs) and molecular mass ranges. Once the spots of interest were isolated, they underwent tryptic digestion followed by liquid chromatography coupled with tandem mass spectrometry (LC-MS/MS) and matrix-assisted laser desorption and ionization-time of flight (MALDI-TOF) sequencing to provide a more comprehensive characterization of the enzyme [32]. It should be noted that 2-DGE might cause protein loss in the spots, so an SDS-PAGE could be applied in parallel to confirm the results [33]. Additionally, this technique encounters difficulties in separating and identifying proteins with extreme characteristics, for example, the smallest, the largest (in terms of mass), the highly acidic, and the highly basic proteins [34]. In this sense, 2-DGE might not be a suitable technique for analyzing the scorpion venom’s lowest migrating fraction in SDS-PAGE gels encompassing neurotoxins, as they are similar in mass (3–8 kDa mol. mass, particularly in buthid venoms) and are mostly basic components, with pIs ≥ 8 in the case of sodium channel-active toxins. In this sense, acid-urea gel electrophoresis using 5% (*v*/*v*) acetic acid as running buffer has provided a better resolution for low mol. mass components in the case of crude venoms and gel filtration chromatography fractions containing sodium- and potassium-channel active neurotoxins from venoms of the genus *Tityus* [35,36,37].

### 2.2. Chromatographic Separation Techniques

#### 2.2.1. Reverse-Phase High-Performance Liquid Chromatography (RP-HPLC)

RP-HPLC is the optimal method for the separation of various scorpion venom components, especially peptides, due to its well established reputation for high-resolution. The fractionation of scorpion venom using RP-HPLC involves the use of two mobile phases with different polarities: an aqueous and an organic mobile phase. The mobile phase is applied to the column at a specific linear or segmented gradient to allow the purification of the target molecule. For scorpion venom separations, the most used acids are trifluoroacetic acid or acetic acid at 0.01% (*v*/*v*) [38]. Segmented gradients are often used in scorpion venom separations since they provide better resolution of small peptidic and non-peptidic components. For instance, for investigating non-peptide small molecule venom constituents in *Hormurus waigiensis*, crude venom was separated by RP-HPLC using this gradient technique: a linear gradient of 0–60% acetonitrile in 0.045%TFA for 120 min, 60–90% acetonitrile buffer for 5 min, 90% acetonitrile buffer for 10 min then 90–0% for 5 min, allowing collection of ~10 peaks, corresponding to glutamic acid, aspartic acid, adenosine, adenosine monophosphate, citric acid and inosine [39]. The most used columns for scorpion venom separations using RP-HPLC include C4, C8 or C18 columns [40,41,42]. The selection of a column depends on the specific objective of the study. For the separation of small peptides, C18 is the preferred column due to its extensive surface area, which allows for broader separation albeit at a slower retention time. This column type is commonly employed for protein purification from scorpion venoms. On the other hand, C8 columns yield sharper peaks and faster retention times, making them suitable for less complex mixtures. However, they may not be ideal for highly complex sample compositions [43]. The collected peaks reflect the partition coefficient of each compound, thus allowing the separation of molecules with close or similar molecular masses. The resulting peaks appear sharper and more defined, with easy time-saving recovery, allowing analytical protein and peptide purification [44,45].

RP-HPLC can be used to isolate and purify peptidic and protein components of scorpion venom for further characterization. The technique started to be used for buthid scorpion venom fractionation in the 1980–90s for the isolation of components responsible for mammalian and insect neurotoxicity from the Brazilian *Tityus serrulatus* and the North African species *Androctonus mauretanicus* and *Androctonus australis* [46,47,48]. RP-HPLC continues to be utilized for the venom characterization of new medically important species, such as *Centruroides hirsutipalpus*, for which venom chromatographic fractions were later subjected to Edman degradation for elucidation of the primary structure of purified components [49]. Fractionation by RP-HPLC was also applied to isolate an anti-viral scorpion component, a Scorpine-like peptide (Smp76), from *Scorpio maurus palmatus* venom [50]. A sequence of RP-HPLCs, starting with the separation of the total crude venom with a C4 semi-preparative column and then a C18 analytical column, was applied to resolve toxic fractions from *Liocheles australasiae* venom. Additionally, the resulting toxic fractions were further separated using a C18 microbore column for the isolation of a single, novel insecticidal peptide (LaIT3) [38]. To maximize the purification process, RP-HPLC can be complemented with other techniques. In certain cases, it may be utilized as the final step to achieve the highest level of purification, as demonstrated in the isolation of neurotoxic peptides from Iranian *Hemiscorpius lepturus* venom and the isolation of a nontoxic peptide (Bot33) from *Buthus occitanus tunetanus* [26,51]. Most commonly, during scorpion venom analysis, RP-HPLC is followed by a mass spectrometric analysis of the fractions. As seen in the case of *Centruroides hirsutipalpus* and *Palamneus gravimanus* venoms, separated fractions by the C18 reverse-phase column were identified by their molecular mass using mass spectrometry [49,52]. 

#### 2.2.2. Size-Exclusion Chromatography (SEC)

SEC is a chromatographic technique that allows the separation of molecules based on their size. The matrix of size exclusion columns is formed of beads of distinct pore sizes. Therefore, depending on the target protein to be isolated, the bead and pore size should be carefully chosen. Different column matrices can be used for SEC, such as dextran polymers (Sephadex), agarose (Sepharose) or polyacrylamide (Sephacryl). SEC could be used as a first step in the process of isolating scorpion proteins from the rest of the venom mixture. For Buthidae α-toxin purification (a scorpion neurotoxin class that alters the inactivation mechanism of voltage-gated sodium channels), SEC was used as a first step to separate venom proteins into fractions of a narrower molecular mass range. This technique is usually complemented by other chromatographic techniques due to its limited resolution. RP-HPLC could be used to further purify the resulting SEC peaks before mass spectrometric analysis [53]. For example, in order to obtain and purify the fraction with the highest glioblastoma growth suppression effect (chlorotoxin-like activity), *Androctonus australis* venom was firstly run on a Sephadex G-50 column; fractions were further separated by a Resolution S cation exchange column, and thirdly, the selected fraction was run on a C18 column [54]. Another example showed the purification of AnCra1 toxin from the Turkish scorpion *Androctonus crassicauda*; nine fractions were obtained by SEC separation using a Sephadex G-50 column. Fraction 5, scoring the highest mammalian toxicity, was then chosen to be further separated. Thus, fraction 5 underwent two additional fractionations by RP-HPLC on a C18 column to obtain the finally purified toxin, AnCra1. The purified peptide was later subjected to Edman degradation, followed by a MALDI-TOF/TOF analysis to determine the primary structure and molecular mass [55]. SEC can also be applied as a single separation method, allowing the collection of different protein fractions, as shown in the case of components altering coagulation parameters from the venom of *Hemiscorpius lepturus*, or exhibiting antimicrobial activities as in the case of *Androctonus australis* [56,57].

Since RP-HPLC and SEC were the most applied techniques for scorpion AMPs separation, Table 1 summarizes the advantages and disadvantages of each technique, along with examples of isolated AMPs.

#### 2.2.3. Ion Exchange Chromatography (IEX)

IEX is a commonly used high-resolution separation technique for scorpion venom components that could be used as a single step or included as a step in a workflow. This method separates molecules based on their charge. The ion exchanger stationary phase could be either cationic or anionic and favors the binding of oppositely charged molecules. The first eluted proteins are the ones with the weakest ionic interaction, followed by the elution of proteins with higher ionic strength simultaneously with the increase in salt concentration. In the case of scorpion venoms, IEX is mainly applied using a linear salt gradient of sodium chloride or ammonium bicarbonate. The choosing of chromatographic columns varies depending on the charge of the protein that needs purification; for cations, cationic exchange chromatography (CEX) is applied, whereas, for anions, anionic exchange chromatography (AEX) is used. Two categories of each IEX exist: strong IEX and weak IEX; weak IEX provides a broader selectivity, making it more favorable for scorpion protein and peptide separation [64]. 

Table 2 shows examples of the implementation of different IEX column types to separate scorpion peptides. Previously, IEX was used as a single fractionation step to isolate a target protein [65]. Nowadays, IEX is more frequently used in bio-guided fractionation with continuous chromatographic separation techniques in order to purify venom proteins and peptides with an improved resolution [66,67]. For example, SEC has been followed by IEX using a diethylaminoethyl (DEAE)-cellulose column to purify a fraction enriched in hyaluronidase activity from *Palamneus gravimanus* venom. Alternatively, it can be followed by two IEX steps: an AEX and a CEX. IEX can also be applied as an intermediate separation step, for instance, between an SEC and an RP-HPLC, to identify the toxic fractions, simplifying, therefore, the research for envenomation antibodies [66,68].

#### 2.2.4. Affinity Chromatography

Affinity chromatography is a highly used technique for the separation of proteins based on the specific, reversible interaction between a molecule and its specific ligand. Due to its simplicity of use, high productivity, and superior precision, this method is considered the optimal choice for scorpion recombinant protein purification. Different types of resins are used for scorpion proteins purification; in particular, nickel chelated resin (Ni) and glutathione S-transferase (GST) resins for recombinant protein purification have been utilized. For example, in order to circumvent the low yield of production for the recombinant insect-selective neurotoxin BjαIT from *Hottentotta judaicus*, a study resorted to applying affinity chromatography using a nitriloacetic (Ni-NTA) agarose column as the purification method, highlighting the high efficiency of this technique for purification of the desired neurotoxin responsible for an insecticidal effect [73]. The same goes for recombinant α-toxin AnCra1, purified from *Androctonus crassicauda*, where affinity chromatography was used for optimal purification [74]. To purify a recombinant GST fusion protein containing a long-chain potassium scorpion toxin from *Mesobuthus martensii* (BmTXKβ), a GST resin (glutathione-sepharose 4B) was used for affinity chromatography. The isolated recombinant protein (GST-rBmTXKβ) was subjected to further testing, confirming its impact on the duration of rabbit myocyte action potential [75]. Streptavidin resins can also serve as a single-step purification method for recombinant peptide antibodies. For example, it was applied with the intent to isolate a fusion protein capable of detecting three *Androctonus australis* sodium-channel toxins associated with the scorpion envenomation syndrome [76]. Affinity chromatography can be accompanied by additional chromatographic separation techniques to achieve more homogenous purification, as seen in the case of a recombinant analgesic peptide from *Mesobuthus martensii*, which required a first immobilized metal ion affinity chromatography (IMAC) using a Nickel-chelating resin, followed by CEX as a second purification step [77]. Thus, nowadays, affinity chromatography is a crucial technique associated with scorpion venom analyses, which has been instrumental in increasing the yield of available target molecules for further research and/or applications. 

Nano-HPLC is another innovative chromatographic technique used for the separation of scorpion venom components [78]. The use of this technique allows for a considerable reduction in the amount of sample to be analyzed and the buffer needed for the separation. This technique is commonly employed in coupling mode with mass spectrometry, which enables precise mass analysis of the separated molecules and enhances ionization efficiency, leading to increased signal intensities [79]. The advantage of this technique lies in the reliability of the results achieved by handling a minimal volume of the scorpion venom sample. This is particularly beneficial given the limited quantities of venom available, considering the typically low yields obtained from live scorpions through electrical stimulation [80]. However, nano-RPLC is most often used in venom analytical processes where there is no preliminary fractionation step, and the raw venom digests are analyzed directly [81].

## 3. Implementation of Separation Methods for Scorpion Venoms

### 3.1. Bioassay-Guided Fractionation 

Bioassay-guided fractionation involves the separation of the crude scorpion venoms primarily through chromatographic techniques. The resulting fractions are subjected to the desired functional assay to identify the active fractions of interest. Eventually, an additional fractionation step might be required to obtain a pure active fraction [82,83]. Depending on the characteristics of the target molecule, different biological assays might be used. Scorpion venom could be submitted to a bioassay-guided fractionation in the pursuit of isolating biomolecules of interest. The newly discovered and purified molecules from scorpion venom can be thus used for drug design owing to their beneficial pharmacological potential. A bioassay-guided fractionation of *Hemiscorpius lepturus* scorpion venom allowed the purification of a novel anticancer protein known as Leptulipin. The process first included fractionation by SEC, using a Sephadex G-50 column, followed by purification of the fraction with the highest anticancer activity using a C18 RP-HPLC column. The final isolated fraction with distinctive cytotoxic activity was then identified by 2-DGE. The bioactivities of *Hemiscorpius lepturus* peptides discovered in specific fractions constitute novel biomolecules with potential pharmacological use [84]. Different neurotoxins in scorpion venom are accountable for the various envenomation effects in humans, especially cardiotoxic peptides that cause severe cardiorespiratory complications to envenomation victims [85]. To identify the cardiotoxic component of *Hemiscorpuis lepturus* venom, bioassay-guided fractionation was employed. SEC using Sephadex G-50 was applied, and fractionation was followed by measurement of optical density, allowing the collection of six peaks. Subsequently, protein profiles were acquired by subjecting the six resulting peaks to 12% SDS-PAGE, which were used to measure levels of specific biochemical cardiac-related enzymes after injection of each fraction. Fraction IV and the whole crude venom were selected for having the highest levels of cardiotoxicity and were then subjected to a histopathological examination of damaged heart tissues. The series of bioassays identified fraction IV of *Hemiscorpius lepturus*, containing low molecular mass peptides, as the fraction responsible for the cardiotoxic effects [86]. Aside from the highly studied neurotoxins, non-disulfide-bridged peptides (NDBPs) were recently put under the spotlight for their important pharmacological activities. In this context, three Iranian scorpion species were subjected to separation by RP-HPLC using C18 columns to isolate fractions with bradykinin potentiating effect. The collected fractions were evaluated for their smooth muscle contracting ability on the guinea pig ileum and rat uterus. Thus, results confirmed the presence of bradykinin potentiating peptides in the venom of the three scorpions, illustrating the use of the above-mentioned fractionation scheme for making such peptides available in the quantities needed for further molecular and structural studies [87]. Interestingly, the implementation of bioassay-guided fractionation has underscored its effectiveness in isolating and characterizing biologically significant compounds present in scorpion venom. Figure 1 presents a selection of essential bioassays employed to isolate and purify molecules with noteworthy biological activity. Given the known presence of analgesic compounds in scorpion venoms, the analgesic assay conducted on mice proved instrumental in assessing the analgesic potential of fractions obtained from *Mesobuthus martensii* venom through a sequence of five successive chromatographic separations. Consequently, this systematic approach facilitated the identification of active analgesic fractions, ultimately leading to identification of the BmKAGAP-SYPU2 component. Similarly, diverse assays such as insect toxicity evaluations contributed to the understanding of the mechanism of action of β-insect depressant toxins derived from the venom of *Isometrus maculatus*, which underwent purification via a two-stage RP-HPLC process. Conversely, a murine toxicity assay applied to fractions derived from *Androctonus australis* venom through four consecutive chromatographic steps led to the identification of AaTx1, a potassium channel blocker belonging to the alpha-KTx family [88,89,90]. Table 3 displays some examples of bioassay-guided fractionation of different peptides contained in scorpion venom using modern separation techniques. In a recent study, transcriptomic analysis through RNA-seq followed by sequence assembly and search in BLAST provided annotation regarding the studied peptide structure contained in *Liocheles australasiae* venom. Bioassay-guided fractionation followed, in order to isolate and characterize venom peptides exhibiting biological activity. First, *Liocheles australasiae* venom was separated by RP-HPLC using a C4 column, while each obtained fraction was subjected to an anti-viral activity test against hepatitis C virus, which allowed identification of the fraction of interest. Further RP-HPLC purification of this fraction using a C18 column was performed. The resulting fractions were also tested for anti-viral activity, and the eluted fraction with the highest activity was identified as containing phospholipase A2 (LaPLA2-1). This process allowed isolation of a newly detected phospholipase A2 in scorpion venom and its characterization for the first time [91]. In the framework of bioassay-guided fractionation, obtaining credible biological results remains the crucial and indispensable element for the success of this experimental approach. This method is not currently widespread compared to conventional analytical techniques. Some limitations and technical difficulties may be at the origin of its slow progress in the field of identification of drug candidates from natural extracts, specifically from scorpion venoms.

### 3.2. Whole Proteome Characterization 

In the quest to analyze the complete composition of various venoms, “omics” technologies have been employed. The advent of these state-of-the-art technologies has facilitated the decomplexation of venom’s composition and the discovery of hidden biological effects. Additionally, they have become crucial tools in the development of antivenoms, specifically targeting the most toxic components found in animal venoms [99]. The large-scale proteomic investigation of venoms is currently known as venomics. This branch of proteomics helps understand venom’s evolution and diversity, facilitating, therefore, the profiling and characterization of venoms [100]. To date, despite the discovery of approximately 2700 scorpion species, the percentage of manually annotated (Swiss-Prot) venom proteins constitute only 14.9% of all the scorpion venom-derived peptide and protein recorded in the Uni-Prot database. This highlights the need for more efficient methods of venom analysis in order to gain more knowledge on scorpion venom components. In scorpion venomics, the transcriptomic approach has played a crucial role in deciphering the expression levels of individual components produced within venom glands, therefore guiding the identification of novel structures and functions of venom proteins and peptides [101,102]. Recently, a new transcriptomic strategy has emerged, focusing on genomics analysis that is still in its early stages and requires further development and expansion [103]. Genomic analysis of scorpion venom now includes full genome sequencing by implementing high-throughput sequencing techniques named Next Generation Sequencing (NGS), revealing the genes encoding venom proteins in a more efficient manner while also enabling the detection of low-abundance components [82]. The implementation of new high-throughput technologies such as 454 pyrosequencing and Illumina sequencing has allowed quantitative and qualitative inter-specific comparisons and also unveiled differential mechanisms of toxin evolution between species [104,105]. The combination of transcriptomic and proteomic approaches increased the protein and peptide coverage in scorpion venom analysis and gave insights about protein composition from the gene sequences identified by transcriptomic analysis [106,107,108]. Collectively, these approaches provide a more comprehensive characterization of scorpion venoms, thereby enhancing the clinical assessment of scorpion envenomation [102,109]. Proteomics has emerged as a key strategy for analyzing scorpion venom, offering a fundamental breakdown of the complex mixture and enabling the profiling and characterization of its components [110,111]. Two approaches can be employed to perform proteomic analysis: bottom-up (BU) and top-down (TD). Unlike BU, where protease-digested proteins are analyzed, in the TD analysis, intact proteins are studied. In recent years, a novel approach known as the middle-down strategy has emerged as a middle ground between bottom-up (BU) and top-down (TD) approaches. The BU approach involves a crucial digestion step, generally using trypsin, to simplify the identification process. In contrast, the middle-down strategy offers a compromise by retaining larger peptide fragments for analysis. These digested peptides are subsequently identified using tandem mass spectrometry (MS/MS) [112]. The implementation of high-throughput mass spectrometry for sequencing enhanced venom analysis, covering a larger number of protein sequences in a faster time, making LC-MS/MS the most applied approach in BU analysis [113,114]. The inclusion of a pre-fractionation/decomplexation step prior to digestion is crucial in facilitating the identification of proteins. This step allows for the detection of low-abundant proteins that are often overlooked, thereby improving the overall protein identification process [115]. Only in shotgun BU analysis a decomplexation step prior to protein digestion is not required, avoiding the possible loss of peptides. Other BU workflows depend on a separation step prior to protein digestion. Both gel-based and chromatographic separation approaches might be used [116].

Workflow 1 (Figure 2) involves the separation of crude scorpion venom using RP-HPLC followed by in-solution trypsin digestion of the obtained fractions. The resulting digested peptides are then subjected to LC-MS/MS analysis. This approach is commonly used particularly for comparing scorpion venom composition. For instance, a study compared venoms from *Chactas reticulatus*, *Opisthacanthus elatus*, *Centruroides edwardsii*, and *Tityus asthenes*. It was observed that compounds from *Centruroides edwardsii* and *Tityus asthenes* eluted below 38% acetonitrile, while those from *Chactas reticulatus* and *Opisthacanthus elatus* eluted at higher acetonitrile percentages (50% and 60%, respectively). This revealed similarities among Buthidae scorpion venoms and differences in comparison to non-Buthidae scorpion venoms. However, all four venoms contained potassium channel-active and sodium channel-active neurotoxins, AMPs, metalloproteinase-like proteins, and phospholipase-like proteins [19]. Another example involves the determination of primary structures of four inhibitory proteins (NaTx-22, NaTx-4, NaTx-36, and NaTx-13) from *Centruroides sculpturatus* venom. These proteins were found to specifically inhibit a sodium channel isoform (Nav 1.8) associated with inflammation and nociception [71]. This workflow is favorable for detecting low-abundance peptides [117]. In workflow 2, the crude venom is separated by a gel-based (SDS-PAGE) technique, followed by in-gel digestion of the separated bands, as shown in Figure 2B. Protein bands marked by Coomassie Blue or silver staining are then analyzed using LC-MS/MS [118]. Prior to gel separation, the use of RP-HPLC on C18 columns can enhance fraction resolution, as demonstrated in the analysis of *Heterometrus longimanus* venom [116,119]. Workflow 3, shown in Figure 2C, involves first a separation of the crude venom using 2-DGE followed by in-gel digestion of protein spots isolated from the gel and the analysis of their protein content by LC-MS/MS. This workflow provides information about peptide mass and pI, enriching the identification of venom proteins [120]. It can be preceded by chromatographic techniques such as SEC. For example, SEC fractions of *Centruroides limpidus* venom were separated by 2-DGE, followed by tryptic digestion of spots and LC-MS/MS analysis. This enabled amino acid sequencing of peptides and a comparison of venoms from female and male scorpions [121]. As previously mentioned, the 2-DGE approach can be utilized for the isolation and identification of specific venom proteins and peptides [32]. One limitation of this approach is the potential loss of proteins during gel separation, making SDS-PAGE a more favorable option. However, despite this limitation, the fourth workflow is generally preferred due to its capability to achieve high proteome coverage. The last workflow (workflow 4) is known as the shotgun approach. In this workflow, scorpion venom is directly digested by trypsin, followed by analysis using a C18 RP-HPLC column and LC-MS/MS identification. This method provides a fast, qualitative analysis that offers a general understanding of venom composition and diversity. For example, when *Serradigitus gertschi* and *Centruroides hentzi* venoms were analyzed using the shotgun approach, 204 and 59 proteins and peptides were identified, respectively [108,122].

In contrast, the TD strategy focuses on analyzing proteins in their intact, native form without a digestion step. This approach overcomes the challenges faced in BU, particularly in terms of misidentification of isoforms and proteoforms. Additionally, it enables the analysis of post-translation modifications, large proteins, and protein-protein interactions [123,124]. Depending on the protein size, TD requires either a denaturation or alkylation step before LC-MS/MS analysis for proteins below 30 kDa or a direct application of native proteins for those above 50 kDa. Despite providing more comprehensive results, this method is less commonly used due to the challenges associated with its execution and the interpretation of the large amount of generated data [124]. Typically, preceding separation methods, particularly RP-HPLC, are employed in the TD process [125]. For example, TD analysis of *Buthus occitanus* venom resulted in the identification of 68 peptides, compared to 36 for in-gel BU and 37 for in-solution BU approaches. To overcome limitations in protein identification, a combination of BU and TD methods has been proposed to ensure a more thorough analysis and a deeper understanding of all venom components. This approach was employed in the analysis of *Buthus occitanus* venom, leading to the identification of a total of 102 proteins [81]. Figure 2 provides a schematic representation of the different workflows.

The emergence of a new analysis concept named middle-down (MD) was influenced by the former two approaches. It includes the digestion of peptides like in BU but with different proteases that generate longer truncated peptides. Following that, the MD approach allows an improvement in protein sequence coverage [126]. Table 4 contains some examples of proteomic workflows adapted for the analysis of different scorpion venoms. To date, the literature on scorpion venom proteomic analysis is scarce, which demands further studies to deepen the knowledge of the use of different venomic approaches to study scorpion venoms.

## 4. Scorpion Venom Antimicrobial Peptides (AMPs)

Recent proteomic analyses have illuminated the highly diversified composition of scorpion venoms, uncovering a link to the realm of AMPs. While these venoms were initially found to contain only small amounts of AMPs, the UniProt database now showcases an impressive collection of over 200 distinct scorpion venom AMPs—a significant advancement from the mere 50 peptides documented a decade ago. This research aims to precisely isolate and characterize these AMPs within scorpion venoms, ultimately unveiling their potential therapeutic uses [127,128,129].

Notably, a significant portion of these identified peptides exhibit remarkable efficacy against multidrug-resistant (MDR) bacteria, which holds great promise in combating the growing threat of antibiotic resistance. Additionally, these scorpion venom-derived AMPs, especially those derived from the Buthidae family, exhibit minimal hemolytic activity, thereby minimizing potential harm to healthy cells [130]. A distinguishing feature of scorpion venom AMPs is their discerning approach to microbial targets. Unlike their counterparts from spider venoms, scorpion venom AMPs display a remarkable specificity towards particular microorganisms rather than exhibiting a broad-spectrum anti-bacterial activity [131]. As seen, three AMPs derived from *Urodacus yaschenkoi* venom recorded high antimicrobial activity against eight MDR bacterial strains, mostly inhibiting *Streptococcus* strains. The minimum bacterial concentration for Uy234, Uy17 and Uy192 against SP10 was 2.9 µM, 23.2 µM and 10.6 µM, respectively, while the minimum bacterial concentration against ST9 was 5.9 µM, 11.6 µM and 15.9 ± 7 µM, respectively. In addition, Uy17 and Uy192 exhibited a lower hemolytic activity (<6%) compared to Uy234 (26.18%) at 380 μM. Despite this, these AMPs displayed a lower hemolytic activity than most scorpion venom AMPs. Collectively, these three peptides exhibit distinct action against MDR bacteria while also showing a low cytotoxic effect. This combination makes them a valuable asset in addressing the increasing prevalence of antibiotic-resistant bacterial strains. Similar findings continue to pave the way for the discovery and development of novel antibiotic-active biomolecules such as Uy17, Uy192, and Uy234 [132]. 

In scorpions, AMPs are short cationic amphipathic peptides divided into three categories according to their structure: (1) the first group contains peptides with cysteine residues and disulfide bridges; (2) the second group lacking cysteine residues contains members with amphipathic α-helix and (3) the third group encompasses members rich in proline and glycine residues. The cysteine-containing AMPs are formed by three to four disulfide bridges. For example, Heteroscorpine-1 (HS-1), from *Heterometrus laoticus* scorpion venom, contains ninety-five amino acids and three disulfide bonds [133]. HS-1 possesses a broad anti-bacterial spectrum, affecting both Gram-negative and Gram-positive strains. The purified fraction of HS-1 scored 300 times higher inhibition activity on *Bacillus subtilis, Klebsiella pneumoniae* and *Pseudomonas aeruginosa* compared to the whole crude venom of *Heterometrus laoticus* [27]. Scorpine is also a scorpion venom-derived AMP from *Pandinus imperator* with three disulfide bonds that constitute only 1.4% of the crude venom. This 75 amino acid-long AMP inhibited both *Bacillus subtilis* and *Klebsiella pneumoniae* strains, recording a MIC of 1 and 10 µM, respectively [62]. 

The non-disulfide bridged AMPs can be long, intermediate or short AMPs. Long-chain non-disulfide bridged AMPs vary in size, with an average of around 40 amino acids. For instance, Hadurin, isolated from *Hoffmannihadrurus aztecus* scorpion venom, is a 41 amino acid-long AMP that constitutes 1.7% of the total venom content. Antimicrobial activity was mainly detected against *Escherichia coli*, *Serratia marscencens* and *Enterococcus cloacae* with MICs lower than 10 µM while the hemolytic activity was significant [63]. Additionally, *Pandinus imperator* venom contains an AMP identified as pandinin-1, which comprises 44 amino acids. The application of pandinin-1 demonstrated notable sensitivity against *Enterococcus faecalis*, *Bacillus subtilis*, *Staphylococcus aureus*, and *Staphylococcus epidermidis*, with minimum inhibitory concentrations (MICs) of 1.3 µM, 5.2 µM, 2.6 µM, and 5.2 µM, respectively. Moreover, pandinin-1 displayed minimal hemolytic effects, with only 1.4% hemolysis observed at the highest concentration tested (44.5 µM). Another polycationic, α-helical peptide was purified from the venom of *Pandinus imperator* venom, pandinin-2, with 24 amino acids, making it an intermediate-chain AMP. This peptide acted mostly on *Enterococcus faecalis, Bacillus subtilis, Staphylococcus aureus* and *Staphylococcus epidermidis* strains with corresponding MICs of 2.4 µM, 4.8 µM, 2.4 µM and 4.8 µM, respectively. Nevertheless, pandinin-2 had a significant hemolytic activity [98].

Intermediate-chain AMPs constitute only 9% of scorpion AMPs described to date. On the other hand, short-chain AMPs are the most commonly found AMPs in scorpion venoms, representing 46% of reported scorpion antimicrobial peptides [133]. Amphipathic peptide CT2 from *Scorpiops tibetanus* venom falls among the short-chain, cationic, non-disulfide bridged AMPs. This AMP has 14 amino acids and inhibits mainly Gram-positive bacteria, especially *Staphylococcus aureus,* with a minimal inhibition concentration (MIC) of 6.25 μg/mL. CT2 was also effective against methicillin-resistant bacterial strains and had a low hemolytic activity even at high concentrations, showing major promise in drug development [134]. A second short-chain AMP, the 13 residue-long cytotoxic linear peptide (IsCT) isolated from *Opisthacanthus madagascariensis* scorpion venom, is also a potential compound for novel antimicrobial drug development. This AMP was significantly more active against Gram-positive bacteria, scoring MICs varying from 0.7 µM to 16.6 µM. Hemolytic activity of IsCT was relatively low, not exceeding 30% at 200 µM [135]. Table 5 summarizes some examples of scorpion venom-derived AMPs, along with their 3D structure retrieved from the UniProt database and their proposed anti-bacterial mechanism of action. Additionally, Table 6 presents the molecular mass, net charge, length and amino acid sequence of some scorpion AMPs.

## 5. Concluding Remarks

To date, the field of venomics has witnessed significant advancements and discoveries, resulting in a substantial increase in our understanding of scorpion venoms. The integration of “omics” technologies and their variants has provided a more comprehensive and advanced insight into the structural/functional relationships in scorpion toxins and other ancillary compounds. Concerning antimicrobial peptides, scorpion venoms have proven to be a rich source of AMPs with a diverse range of mechanisms of action and structural features. The key to conducting optimal scorpion venom separation analyses lies in the careful selection of separation techniques that align with the study’s objectives. In the case of AMPs, these techniques should be coupled with functional assays to genuinely identify novel compounds. This review presents four alternative workflows for venom analysis. Workflow 1 involves venom chromatographic pre-fractionation followed by the tryptic digestion of selected fractions. In workflow 2, venom components are separated using SDS-PAGE, and selected bands undergo digestion. In workflow 3, venom is resolved using 2-DGE, and selected protein spots are analyzed through LC-MS/MS. Workflow 4 employs a “shotgun” approach, where venom is directly digested by trypsin, followed by RP-HPLC analysis and the LC-MS/MS identification of selected fractions. Each approach has its strengths and weaknesses, highlighting that there is no singular methodology for analyzing all aspects of scorpion venom, particularly considering their species-specific complexity. Despite the remarkable progress in scorpion venomics, a significant gap in our understanding of this field remains. Further structural and functional research is essential to unravel this complexity and expand our knowledge of potential bioactive molecules for therapeutic applications.

## Figures and Tables

**Figure 1 antibiotics-12-01380-f001:**
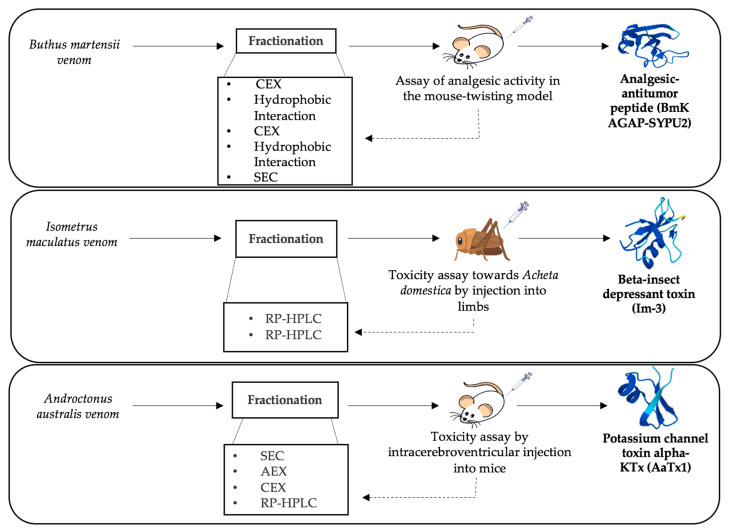
Schematic representation of bioassay-guided fractionation examples for isolation of three scorpion neurotoxins [88,89,90].

**Figure 2 antibiotics-12-01380-f002:**
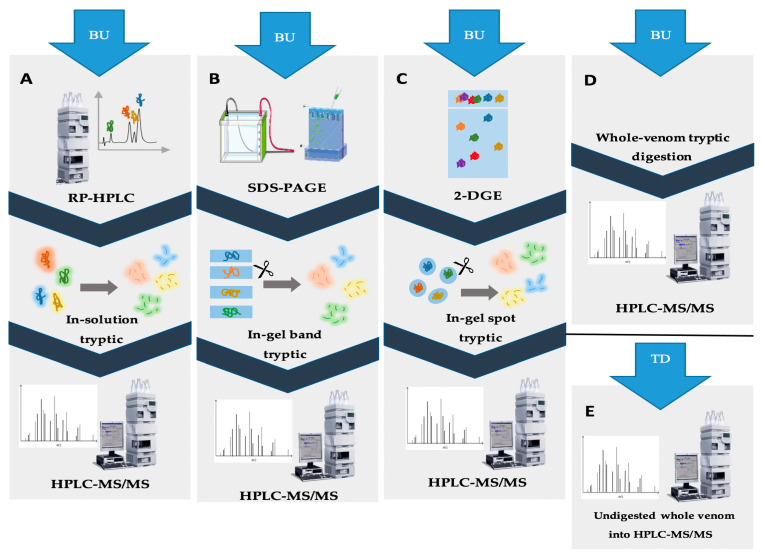
Illustration of various proteomic approaches for scorpion venom analysis. (**A**) The second workflow in the BU approach involves RP-HPLC fractionation, followed by in-solution tryptic digestion of each peak. The resulting digested peptides are then analyzed using HPLC-MS/MS for protein identification. (**B**) The third workflow in the BU approach includes the fractionation of crude venom using SDS-PAGE, followed by tryptic digestion of the protein bands. These bands are subsequently analyzed using HPLC-MS/MS for identification. (**C**) The fourth workflow in the BU approach starts with a first fractionation of crude venom by 2-DGE, followed by spot tryptic digestion. Protein identification is then performed using HPLC-MS/MS. (**D**) The shotgun approach involves direct digestion of crude venom by trypsin, followed by analysis using HPLC-MS/MS. (**E**) The TD approach entails the direct analysis of intact proteins in crude venom using HPLC separation, followed by tandem mass spectrometry (MS/MS). These different approaches provide a range of options for analyzing scorpion venom, allowing researchers to choose the most suitable method based on their specific objectives and requirements.

**Table 1 antibiotics-12-01380-t001:** Separation techniques used for the isolation of some scorpion venom AMPs with advantages and disadvantages.

Separation Techniques	Advantages	Disadvantages	Example of Purified AMP	References
RP-HPLC	- High precision - High sensitivity - High purity	- Costly - Complex	Vejovine [11]	[58]
			Cytotoxic linear peptide (IsCT) [59]	
			Scorpine-like peptide (Smp76) [50]	
SEC	- Favorable as first step for separating complex mixture - Fast elution - Good reproducibility	- Overlap of AMPs in a large quantity sample - Time-consuming overall - Expensive - Not suited for tertiary structures	First step in Heteroscorpine-1 purification [27]	[60,61,62]
			First step in Scorpine purification	
			First step in Hadurin purification [63]	

**Table 2 antibiotics-12-01380-t002:** Type of column used in each type of IEX with an example of purified protein using each one.

Type of IEX	Column Name	Scorpion Species	Purified Molecule	References
**Strong AEX**	Quaternary ammonium (Q) column	*Scorpio maurus*	Phospholipase A2 (*Sm*-PLVG)	[69]
**Weak AEX**	Diethylaminoethyl (DEAE) column	*Buthotus schach*	BS311 and BS313	[70]
**Strong CEX**	Sulphopropyl (SP) column	*Centruroides sculpturatus*	Proteins inhibiting Nav1.8	[71]
**Weak CEX**	Carboxymethyl (CM) column	*Mesobuthus martensii*	Scorpion venom peptide (SVP-B5)	[72]

**Table 3 antibiotics-12-01380-t003:** Different separation strategies used to isolate enzyme components from scorpion venoms. Da: Dalton.

Purified Molecule	Scorpion Species	Molecular Mass (Da)	Separation Process	Column Used	References
**Phospholipase** **A2**	*Liocheles australasiae*	13,079.8	RP-HPLC RP-HPLC LC/MS	C4 C18 C18	[91]
	*Hemiscorpius lepturus*	14,000	SEC RP-HPLC RP-HPLC	Sephadex G-50 Semi preparative C8 Analytical C8	[92]
	*Scorpio maurus*	17,000	SEC AEX Hydrophobic Interaction HPLC	Sephadex G-100 Q-Sepharose Phenyl-Sepharose Nucleogel GFC 300-8	[69]
	*Heterometrus laoticus*	14,018.4	SEC CEX RP-HPLC	Sephadex G-50 CM-650 M C4	[32]
**Hyaluronidase**	*Rhopalurus junceus*	45,000–60,000	SEC	Superdex 75	[93]
	*Tityus serrulatus*	49,312	SEC RP-HPLC	Sephadex G-50 Analytical C8	[94]
	*Palamneus gravimanus*	52,000	SEC IEX SEC	Sephadex G-75 DEAE-cellulose	[52]
**Metalloproteinase**	*Tityus serrulatus*	22,000 24,000	AEX SEC	DEAE Diol-300	[18]
		25,500	SEC RP-HPLC	Sephadex G-50 C18	[19]
**Serine proteinase**	*Mesobuthus martensii*	33,000	SEC CEX RP-HPLC	Superdex G-75 UNO-Q C8	[95]
	*Scorpio maurus*	25,000	CEX SECFP AEXSEC	DEAE-SephadexSephadex G-100SP-SepharoseSephadex G-50	[96]
**Neurotoxins**	*Mesobuthus martensii*	7246.40	CEX Hydrophobic Interaction CEX Hydrophobic Interaction SEC	SP-Sepharose Phenyl Sepharose 4 SP-Sepharose Phenyl Sepharose 4 Superdex Peptide HR 10/30	[88]
	*Centruroides suffusus suffusus*	7524.9 7537.6 7588.6 13,596	RP-HPLC CEX RP-HPLC	C18 TSK-gel sulfopropyl C18	[97]
	*Isometrus maculatus*	6894	RP-HPLC RP-HPLC	C4 C18	[89]
	*Androctonus australis*	3849.5	SEC SEC Exchange FPLC RP-HPLC	Sephadex G-50 Resource S C18	[90]
	*Hemiscorpius lepturus*	4874 5107	SEC AEX CEX RP-HPLC	Sephadex G-50 DEAE-Sepharose CM-Sepharose C8	[26]
**AMPs**	*Heterometrus laoticus (Heteroscorpine-1)*	8293	SEC CEX	Sephadex G-50 CM-Sepharose	[27]
	*Pandinus imperator (Scorpine)*	8350	SEC CEX RP-HPLC	Sephadex G-50 CM-Cellulose C18	[62]
	*Hoffmannihadrurus aztecus (Hadrurin)*	4436	SEC HPLC HPLC	Sephadex G-50 C18 C18	[63]
	*Pandinus imperator (Pandinin-1)*	4799	RP-HPLC CEX RP-HPLC	C18 TSK-gel sulphopropyl C4	[98]

**Table 4 antibiotics-12-01380-t004:** Examples of different workflows and strategies used in scorpion venom proteomic analysis and consequent findings. Da: Dalton.

Workflow	Scorpion Species	No. of Proteins	Main Protein Distribution/Most Abundant Venom Components	References
Workflow 1 ** *Shotgun strategy* **	*Tityus obscurus*	ND	Metalloproteinase (47.48%) NaScTxs (13.80%) KScTxs (11.45%) Conserved venom components (10.26%) AMPs (3.51%) Other proteinases (5.74%) Other components (7.76%)	[109]
	*Tityus serrulatus*	ND	Metalloproteinase (36.55%) NaScTxs (14.19%) KScTxs (15.60%) Conserved venom components (14.99%) Hypotensin (4.91%) Other component (15.98%)	
	*Rhopalurus agamemnon*	230	NaScTxs (16.95%) KScTxs (2.17%) AMPs (1.73%) Housekeeping proteins (40.43%) Metalloproteinase (6.12%) Amylase (2.825%) Others (29.775%)	[118]
** *Combination of workflow 2 + 3 + 4* **	*Mesobuthus martensii*	227	NaScTxs (9.69%) KScTxs (5.32%) AMPs (0.44%) Regulation proteins (11.2%) Structure proteins (7.04%) Metabolism proteins (7.04%) Other components (59.27%)	[33]
** *Combination of workflow* ** ** *1 + 3 + TD* **	*Buthus occitanus*	102	NaScTxs (77%) KScTxs (14%) ClScTxs (3%) CaScTxs (1%) Toxin Acra (1%) Other components (4%)	[81]
** *Combination of workflow 2 + 3 + TD* **	*Tityus serrulatus*	147	KScTxs (12.19%) NaScTxs (10.81%) Enzymes (32%) AMPs (2%) Other components (43%)	[123]

**Table 5 antibiotics-12-01380-t005:** Selected AMPs isolated from scorpion venom, with their 3D structure retrieved from the UniProt database and their mechanism of action against bacterial strains [136,137]. Labeled in red is the signal part of the structure that is later lost when the protein matures. The N- and C- terminal of the proteins are also represented.

Scorpion Species	AMP	Mechanism of Action	Structure	Reference
*Heterometrus* *laoticus*	Heteroscorpine-1 (HS-1)	Formation of blebs on the membrane	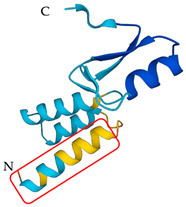	[27]
*Hoffmannihadrurus aztecus*	Hadrurin	Lysis of zwitterionic phospholipids using hydrophobic interactions and acidic liposomes using electrostatic forces	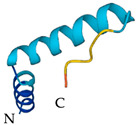	[63]
*Pandinus* *imperator*	Pandinin-1	Membrane disruption and pore formation	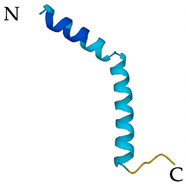	[138]
*Pandinus* *imperator*	Pandinin-2	Liaison with degradation of lipid membrane and pore formation	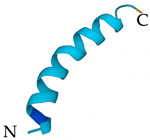	[139]
*Pandinus* *imperator*	Scorpine	Membrane disruption by hydrophobic liaisons and cell penetration	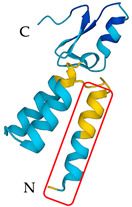	[140]
*Scorpiops* *tibetanus*	Amphipathic peptide CT2	Immediate disruption of bacterial membrane causing rapid killing	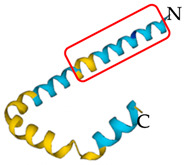	[134]
*Vaejovis* *mexicanus*	Vejovine	Membrane disruption by direct interaction through the N-terminal region	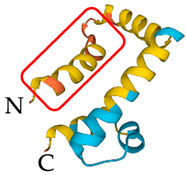	[11]

**Table 6 antibiotics-12-01380-t006:** Molecular masses, net charges, lengths and amino acid sequences of selected scorpion AMPs.

Scorpion AMPs	Molecular Mass	Net Charge	Length	Amino Acid Sequence	Reference
Amphipathic peptide CT2	7930 Da	+2	69	N- MKTQFAVLIISMILMQMLVQTEAGFWGKLWEGVKSAIGKRSLRNQDQFDNMFDSDLSDADLKLLDDLFD -C	[136,137]
Cytotoxic linear peptide IsCT2	1463.92 Da	+2	71	N- MKTQFAILLVALVLFQMFAQSEAIFGAIWNGIKSLFGRRALNNDLDLDGLDELFDGEISQADVDFLKELMR -C	
Hadrurin	4436 Da	+5	41	N- GILDTIKSIASKVWNSKTVQDLKRKGINWVANKLGVSPQAA -C	
Imcroporin	1760 Da	+2	74	N- MKFQYLLAVFLIVLVVTDHCQAFFSLLPSLIGGLVSAIKGRRRRQLEARFEPKQRNFRKRELDFEKLFANMPDY -C	
Meucin-24	2753.95 Da	+4	88	N- MMKQQFFLFLVIVMISSVIEAGRGREFMSNLKEKL SGVKEKMKNSWNRLTSMSEYACPVIEKWCEDHCQAKNAIGRCENTECKCLSK -C	
Meucin-25	3095.56 Da	+4	56	N- MFRIEYSLVQLLLRNVTIPLLLIIQMHIMSSVKLIQIRIWIQYVTVLQMFSMKTKQ -C	
Mucroporin	2031.58 Da	+2	74	N- MKVKFLLAVFLIVLVVTDHCHALFGLIPSLIGGLVSAFKGRRKRQMEARFEPQNRNYRKRELDLEKLFANMPDY -C	
Pandinin-1	4799.2 Da	+1	44	N- GKVWDWIKSAAKKIWSSEPVSQLKGQVLNAAKNYVAEKIGATPT -C	
Pandinin-2	2612.6 Da	+1	24	N- FWGALAKGALKLIPSLFSSFSKKD -C	
Scorpine	8350 Da	+3	94	N- MNSKLTALIFLGLIAIAYCGWINEEKIQKKIDERMGNTVLGGMAKAIVHKMAKNEFQCMANMDMLGNCEKHCQTSGEKGYCHGTKCKCGTPLSY -C	
Vejovine	4873 Da	+4	82	N- MNAKTLFVVFLIGMLVTEQVEAGIWSSIKNLASKAWNSDIGQSLRNKAAGAINKFVADKIGVTPSQAASMTLDEIVDAMYYD -C	
Uy17	1369.43 Da	+2	13	N- ILSAIWSGIKGLL -C	[141]
Uy192	1459.98 Da	+2	13	N- FLSTIWNGIKGLL -C	
Uy234	1986.19 Da	+3	18	N- FPFLLSLIPSAISAIKRL -C	

## Data Availability

Not applicable.

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
