# Peer review of "Scorpion Venom as a Source of Antimicrobial Peptides: Overview of Biomolecule Separation, Analysis and Characterization Methods"

_antibiotics, 2023, doi:10.3390/antibiotics12091380_

Round 1

Reviewer 1 Report

The scope of the journal and the title of the manuscript would make necessary to dedicate a subchapter to the antimicrobial/ antiviral components identified in the venom of various scorpion species, and to have a clear focus on them along all sections of the manuscript (separation methods, ‘Conclusions’). The information should be summarized in two respective tables (one with the antimicrobial peptides and their targets, and another with the methods for separating them, with advantages, disadvantages, limitations).

Line 37 - ‘scorpion proteome’ should be corrected to ‘scorpion venom proteome’

2.2. The limitations of SDS-PAGE would rather fit at the end of section 2.1.

2.2.1 The technical details are too extensive

e.g. 'It is essential to emphasize that no single definitive workflow exists,' line 38

Author Response

The scope of the journal and the title of the manuscript would make necessary to dedicate a subchapter to the antimicrobial/ antiviral components identified in the venom of various scorpion species, and to have a clear focus on them along all sections of the manuscript (separation methods, ‘Conclusions’). The information should be summarized in two respective tables (one with the antimicrobial peptides and their targets, and another with the methods for separating them, with advantages, disadvantages, limitations).

Indeed, we added therefore a subchapter before the conclusion focused on scorpion AMPs with extensive examples. A clear focus on AMPs is now presented in the manuscript (abstract, conclusion, etc…) and two tables were also added accordingly one in the separation methods and one in the AMP section. We also added selected AMP data in Table 3. All changes are highlighted in yellow in the manuscript as recommended.

Concerning antiviral components:

We are currently preparing another study regarding the antiviral effects of animal venoms, including scorpion venoms, and hope to submit it soon. Therefore, with the permission of reviewer 1 author, we prefer not to include these data in this review and keep them for the one currently in preparation.

We thank you very much for your understanding…

Line 37 - ‘scorpion proteome’ should be corrected to ‘scorpion venom proteome’

Done

2.2. The limitations of SDS-PAGE would rather fit at the end of section 2.1.

Done

2.2.1 The technical details are too extensive

Done, we have lightened the technical details as recommended. 

e.g. 'It is essential to emphasize that no single definitive workflow exists,' line 38

Done

We would like to thank reviewer 1 most sincerely for his comments and remarks, which we are sure will help improve our manuscript. 

Reviewer 2 Report

1. General comment

In the manuscript, the author reviewed a comprehensive overview of the key methodologies employed in the extraction, purification, analysis, and characterization of antimicrobial peptides and other bioactive molecules present in scorpion venom.

This review also presents adapted proteomic workflows offering insights into the comprehensive characterization of the entire scorpion proteome.

2. Major revision

1) Line 323~375 and Figure 1

It is strongly recommended to explain Figure 1 in the sentences of line 323~375.

2) Line 384~550, Figure 2 and Table 3

As it is very difficult to understand the relationship among the sentences of line 384~550, Figure 2 and Table 3, it is strongly recommended to explain the question mentioned below.

A) It is strongly recommended to explain Figure 2 in the sentences of line 384~550.

B) It is strongly recommended to unify the writing of workflow described as “second, third or fourth workflow in Figure 2”, “workflow 1, 2, 3 and 4 in Table 3” and “shotgun approaches, second, third or fourth workflow in line 384~550”.

3. Minor revision

1) Line 152: Revise “matrix-sssisted” to matrix-assisted”.

2) Table 3: Revise “Nb. of proteins” to “No. of proteins”.

Author Response

  1. General comment

In the manuscript, the author reviewed a comprehensive overview of the key methodologies employed in the extraction, purification, analysis, and characterization of antimicrobial peptides and other bioactive molecules present in scorpion venom.

This review also presents adapted proteomic workflows offering insights into the comprehensive characterization of the entire scorpion proteome. 

  1. Major revision

1) Line 323~375 and Figure 1

It is strongly recommended to explain Figure 1 in the sentences of line 323~375.

Thank you for your insightful comments and suggestions.

Regarding the first point, we added a paragraph highlighted in yellow, in the new version of our manuscript in lines 364 – 378, that explains Figure 1 in details.

 2) Line 384~550, Figure 2 and Table 3

As it is very difficult to understand the relationship among the sentences of line 384~550, Figure 2 and Table 3, it is strongly recommended to explain the question mentioned below.

Indeed, we agree. We have improved this part of the manuscript according to your suggestion.

  1. A) It is strongly recommended to explain Figure 2 in the sentences of line 384~550.

According to your recommendation, we have added several phrases to the paragraph of “whole proteome characterization” to explain each of the four workflows that are described and shown in Figure 2. The changes are highlighted in the new version of the manuscript.

  1. B) It is strongly recommended to unify the writing of workflow described as “second, third or fourth workflow in Figure 2”, “workflow 1, 2, 3 and 4 in Table 3” and “shotgun approaches, second, third or fourth workflow in line 384~550”.

To fulfill your recommendation we have unified the description of all workflows by numbering them as follows: Workflow 1 (RP-HPLC then in-solution tryptic digestion), Workflow 2 (SDS-PAGE then in-gel protein digestion), Workflow 3 (2DGE then in gel protein digestion) and workflow 4 (Shotgun proteomics). This nomenclature was used in the text and the corresponding panels of figure 2 were assigned also in the text. The nomenclature is also similar to the one used in Table 3 and therefore the description of the proteomic workflows was unified through the text. 

  1. Minor revision

1) Line 152: Revise “matrix-sssisted” to “matrix-assisted”.

Done

2) Table 3: Revise “Nb. of proteins” to “No. of proteins”.

Done

We would like to thank reviewer 2 most sincerely for his comments and remarks, which we are sure will help improve our manuscript. 

Reviewer 3 Report

The work “Scorpion Venom as a Source of Antimicrobial Peptides: Overview of Biomolecule separation, Analysis and Characterization Methods” focuses on the instrumental methods of analysis of the venom composition of scorpions. The valuable information concerning AMP is presented in Table 3 of the manuscript. According to it, from as many as 7 scorpion species, AMP have been found in only single scorpion species,  Tityus obscurus. The relative content of these peptides among venom proteome is 3.2%.

However, if one addresses uniprot.org database, it can be found that now in this database no less than 200 AMP from scorpion venoms are present. 10 years ago no more than 50 peptides can be found in this database. Thus the work is going on to isolate, identify AMP in scorpion venom. Many of these peptides act on multi-drug-resistant bacteria and possess low hemolytic activity. Compared to spider venoms, the scorpion venom AMP feature not broad-spectrum antibacterial activity but act specifically on certain microorganisms. Unfortunately, this information is missing in the manuscript. Although, according to its title, it should be there. 

To avoid the unfulfilled hopes inspired by the title of the work I would recommend the authors:

1. Add a chapter describing AMP from scorpion venoms and their characteristics: length, charge, presence/absence of disulfide bonds, activity spectrum (affected bacteria with indication of MIC, MBC) and hemolytic properties;

2. In the added chapter provide data on the specific action of scorpion AMP on Multi-Drug-Resistant (MDR) microorganisms;

3. Add more figures illustrating the spatial structure of AMP from scorpion venom (in uniprot.org the alpha-fold structures of the all peptides are available) and possibly their structure-function relationships; currently, the article contains only 2 figures and 3 tables; is this mini-review? 

4. Rewrite the Abstract and conclusion sections to reflect the information above and provide answer to the question: “Which workflow is the most useful for identification of AMP in scorpion venom”.

Currently, the article features also the following ambiguities:

1. In Table 3 no percentages are provided for the components of the Tityus discrepans venom. For other venom the sum of the percentages of the components is less than 100%. Other components have not been identified? Please, specify this.

2.  In lines 391-393 the numbers are presented: “To date, despite the discovery of approximately 2700 scorpion species, only 12.1% of the 8989 proteins found in venom glands have been manually annotated.” Thus, 2700 scorpions do use 8989 proteins in their venoms? Thus, in the venom of each scorpion only 3 proteins are present? Probably, this is incorrect. Thus the sentence should be modified to remove ambiguity. Please, specify the size of the known proteomes of the selected scorpion species and the percentage of the identified proteins within each of them.     

A phrase difficult for understanding: "...these combinations have proven instrumental in optimizing the identification of venom components..."

Author Response

The work “Scorpion Venom as a Source of Antimicrobial Peptides: Overview of Biomolecule separation, Analysis and Characterization Methods” focuses on the instrumental methods of analysis of the venom composition of scorpions. The valuable information concerning AMP is presented in Table 3 of the manuscript. According to it, from as many as 7 scorpion species, AMP have been found in only single scorpion species,  Tityus obscurus. The relative content of these peptides among venom proteome is 3.2%.

However, if one addresses uniprot.org database, it can be found that now in this database no less than 200 AMP from scorpion venoms are present. 10 years ago no more than 50 peptides can be found in this database. Thus the work is going on to isolate, identify AMP in scorpion venom. Many of these peptides act on multi-drug-resistant bacteria and possess low hemolytic activity. Compared to spider venoms, the scorpion venom AMP feature not broad-spectrum antibacterial activity but act specifically on certain microorganisms. Unfortunately, this information is missing in the manuscript. Although, according to its title, it should be there. 

To avoid the unfulfilled hopes inspired by the title of the work I would recommend the authors:

  1. Add a chapter describing AMP from scorpion venoms and their characteristics: length, charge, presence/absence of disulfide bonds, activity spectrum (affected bacteria with indication of MIC, MBC) and hemolytic properties;
  2. In the added chapter provide data on the specific action of scorpion AMP on Multi-Drug-Resistant (MDR) microorganisms;
  3. Add more figures illustrating the spatial structure of AMP from scorpion venom (in uniprot.org the alpha-fold structures of the all peptides are available) and possibly their structure-function relationships; currently, the article contains only 2 figures and 3 tables; is this mini-review? 

We took into consideration your recommendations and we added a subchapter in the new version of our manuscript (see point 4, page 17) discussing AMPs in a more focused way, describing the categories their length, charge, MIC, effect on MDR … We also added a Table 5 containing AMPs isolated from scorpion venom with their mechanism of action and 3D structure.

  1. Rewrite the Abstract and conclusion sections to reflect the information above and provide answer to the question: “Which workflow is the most useful for identification of AMP in scorpion venom”.

Done, the abstract and conclusion of the review have been thoroughly revised and improved as recommended by reviewer 3.

Currently, the article features also the following ambiguities:

  1. In Table 3 no percentages are provided for the components of the Tityus discrepans venom. For other venom the sum of the percentages of the components is less than 100%. Other components have not been identified? Please, specify this.

We first wanted to only put percentages for the main components, but we fixed the mistake and filled the table. Concerning Tityus discrepans venom we couldn’t find the percentages, so we removed it. We also added AMP percentages.

  1. In lines 391-393 the numbers are presented: “To date, despite the discovery of approximately 2700 scorpion species, only 12.1% of the 8989 proteins found in venom glands have been manually annotated.” Thus, 2700 scorpions do use 8989 proteins in their venoms? Thus, in the venom of each scorpion only 3 proteins are present? Probably, this is incorrect. Thus the sentence should be modified to remove ambiguity. Please, specify the size of the known proteomes of the selected scorpion species and the percentage of the identified proteins within each of them. 

We modified the phrase for better understanding and corrected the numbers after checking Unilrot updated list. Just to clarify, we are talking that the percentage of manually annotated proteins (in Swiss-Prot) is still small compared to the predicted proteins from TrEMBL. Thus, eliciting the need for more proteomic approaches to manually annotate the rest of proteins.

 A phrase difficult for understanding: "...these combinations have proven instrumental in optimizing the identification of venom components..."

We reconstructed the phrase for better understanding and we highlighted changes.

We would like to thank reviewer 3 most sincerely for his comments and remarks, which we are sure will help improve our manuscript. 

Round 2

Reviewer 1 Report

The manuscript was substantially updated, according to our comments.

Careful editing of the English language is recommended

Author Response

The manuscript was substantially updated, according to our comments.

Careful editing of the English language is recommended.

English corrections have been made throughout the manuscript (highlighted in green with track changes).

Many thanks to Reviewer 1 for all his comments during Round 1 and Round 2, which ultimately contributed to the production of a greatly improved version of our review.

Reviewer 3 Report

Concerning Table 5, the presented structure are dificult to understand. Please, mark N- and C-termini. It seems, the structure encompasses both signal and mature part. Please, remove the signal part, because only the mature part is active. Also, some peptides are mentioned in the text with no additional information about them (at least, the amino acid sequences), e.g. Uy234, 17, 192. These peptides have not been mentioned in the review [133]. Probably, a separate table with amino acid sequences of the mentioned in the text peptides is worth to be presented (the amino acid sequences of the peptides from Table 5 also can be presented in this additional table). May be, this additional Table could contain information on the net electrical charge, length, mass etc of the peptides (see e.g. Table 1 in the review [133]).

And, finally, I recommend authors to discuss the new peptides, not discussed in ref. [133].

Some sentences are still difficult to understand, e.g. "This cationic AMP length 14 amino acids and exhibits.." (line 641). Editing is required.

Some sentences are still difficult to understand, e.g. "This cationic AMP length 14 amino acids and exhibits.." (line 641). Editing is required!

Author Response

Concerning Table 5, the presented structure are dificult to understand. Please, mark N- and C-termini. It seems, the structure encompasses both signal and mature part. Please, remove the signal part, because only the mature part is active. Also, some peptides are mentioned in the text with no additional information about them (at least, the amino acid sequences), e.g. Uy234, 17, 192. These peptides have not been mentioned in the review [133]. Probably, a separate table with amino acid sequences of the mentioned in the text peptides is worth to be presented (the amino acid sequences of the peptides from Table 5 also can be presented in this additional table). May be, this additional Table could contain information on the net electrical charge, length, mass etc of the peptides (see e.g. Table 1 in the review [133]).

Answer : Following your request, the N and C terminal were marked in green and purple on the 3D structures in Table 5. Also the structures containing a signal part were marked in red on the same structure taken from UniProt.

We also made a Table 6 containing all the AMPs mentioned in the manuscript. The table holds the length, mass, sequence and charge of the scorpion AMPs.

See highlight green in the manuscript, Thanks

 And, finally, I recommend authors to discuss the new peptides, not discussed in ref. [133].

Answer: We took your recommendation into consideration and added a marked discussion in green (line 704 in the last version of our manuscript).

Some sentences are still difficult to understand, e.g. "This cationic AMP length 14 amino acids and exhibits.." (line 641). Editing is required.

Answer : Correction were made accordingly and the sentence was changed and marked in green in the manuscript.

We thank Reviewer 3 for all his comments during Round 1 and Round2, which ultimately contributed to the production of a greatly improved version of our review.  
